# Controlling Nutritional Status (CONUT) Score as a Predictor of Prognosis in Non-Small Cell Lung Cancer

**DOI:** 10.3390/nu17213416

**Published:** 2025-10-30

**Authors:** Raffaella Pagliaro, Luca Scalfi, Ilaria Di Fiore, Anna Leoni, Umberto Masi, Vito D’Agnano, Carmine Picone, Filippo Scialò, Fabio Perrotta, Andrea Bianco

**Affiliations:** 1Department of Translational Medical Sciences, University of Campania “L.Vanvitelli”, 80131 Naples, Italy; ilaria.difiore@studenti.unicampania.it (I.D.F.); anna.leoni@studenti.unicampania.it (A.L.); umberto.masi@studenti.unicampania.it (U.M.); vito.dagnano@studenti.unicampania.it (V.D.); fabio.perrotta@unicampania.it (F.P.); andrea.bianco@unicampania.it (A.B.); 2Clinic of Respiratory Diseases “L. Vanvitelli”, A.O. dei Colli, Monaldi Hospital, 80131 Naples, Italy; 3Department of Public Health, School of Medicine, University of Naples Federico II, 80131 Naples, Italy; luca.scalfi@unina.it; 4Division of Radiology, Istituto Nazionale Tumori IRCCS Fondazione Pascale-IRCCS di Napoli, 80131 Napoli, Italy; carminepico@libero.it; 5Department of Medicine and Health Science, Vincenzo Tiberio University of Molise, 86100 Campobasso, Italy; 6Department of Molecular Medicine and Medical Biotechnologies, University of Naples Federico II, 80131 Naples, Italy; filippo.scialo@unina.it; 7CEINGE-Biotecnologie Avanzate Franco Salvatore, Via G. Salvatore 486, 80145 Naples, Italy

**Keywords:** malnutrition, CONUT score, non-small cell lung cancer, survival outcomes, treatment response, nutritional index

## Abstract

*Background:* Several studies have reported the association between malnutrition risk and survival outcomes in surgically treated lung cancer patients; in contrast, limited data are available on patients undergoing medical therapies. The aim of this study was to evaluate malnutrition risk (using the CONUT score) and its relationships with treatment response and survival in patients with an advanced stage of non-small cell lung cancer (NSCLC). *Methods:* We retrospectively evaluated 109 patients with a diagnosis of locally advanced and advanced NSCLC. Patients were assessed at baseline (before any therapy), after four cycles of therapy, and during a follow-up period of 4 years. *Results:* Thirty-two (29.4%) NSCLC patients had a CONUT score ≥ 2. Patients with a higher CONUT score were more likely to have brain metastases while patients with a CONUT score < 2 had a significantly better response to therapy in terms of partial response/stable disease. In addition, The Kaplan–Meier curves revealed a significantly worse prognosis for the high CONUT than the low CONUT group regarding both OS and PFS. Cox regression analysis indicated that the CONUT score (≥2) was as a significant determinant of OS and PFS in the patients studied even after considering other possible predictors. *Conclusion:* This study suggests that malnutrition risk assessment with CONUT score may be valuable in the prognosis assessment of advanced-stage NSCLC patients.

## 1. Introduction

Disease-related malnutrition/undernutrition (from here on, malnutrition), which is a clinical syndrome due to deficiencies of energy and/or nutrients, is characterized by alterations in both body composition and body functions due to energy and is negatively related to various health outcomes [1,2]. Malnutrition is often found in cancer patients, at the time of diagnosis or throughout the course of disease, causing increased morbidity and mortality [3]; its prevalence and incidence depend on the type and stage of disease, as well as clinical settings and treatments [2]. In such a case, unintentional weight loss, a major indicator of malnutrition, may be ascribed to the interactions of several factors such as reduced food intake due to local effects of cancer, side effects of treatments or complications such as infections and toxicity, depression, poor mental status, etc. [4]; it is commonly associated with poor prognosis, worse quality of life (QoL), low physical activity level, adverse effects due to treatment and an impaired response to treatment [2,5]. Further determinants of malnutrition may be immobilization, anorexia, oral and intestinal mucositis with dysphagia, diarrhea and poor dentition [6,7]. In addition, tumour-induced inflammation and catabolism may lead to muscle wasting and decreased fat-free mass (FFM) [8].

There is a widespread consensus on the fact that it is mandatory to create and enforce regular protocols for identifying malnourished cancer patients and to ensure that timely nutritional support is provided [4]. To avoid underdiagnosing and undertreating, all cancer patients should be screened for malnutrition risk as a first step in the nutritional care process [7], while nutritional assessment is subsequently used to examine in depth nutrition-related problems and their causes [9]. Various screening tools have been developed to identify patients who are at risk of malnutrition [10,11]. For instance, the Guideline of the European Society for Clinical Nutrition and Metabolism (ESPEN) recommends the use of Nutritional Risk Screening (NRS), Malnutrition Universal Screening Tool (MUST) and Malnutrition Screening Tool (MST) [12,13]. More recently, screening tools including hematobiochemical parameters have also been widely used in cancer patients such as the Controlling Nutritional Status (CONUT) score [14]. The CONUT score provides healthcare professionals with a quantitative assessment of the malnutrition risk and aids in identifying those at a higher risk of malnutrition-related complications [11]. It categorizes patients into different risk groups based on a cumulative score based on serum albumin level, total lymphocyte counts and total cholesterol level [11]. Overall, the CONUT score is thought to be influenced by poor nutritional status as well as inflammation and immune system suppression.

A low serum albumin is indicative of malnutrition and inflammation, which are both common in cancer patients, while a reduced lymphocyte count might reflect immune system suppression [15].

Lung cancer (LC) is a malignant tumour that negatively impacts nutritional status due to various factors, including the disease itself and the side effects of treatments like chemotherapy, radiation therapy, and surgery [16]. The prevalence of malnutrition is reported to range from 35 to 75%, depending on the stage of disease, type of treatment and assessment method used [17]. However, the malnutrition is associated with negative outcomes, including impaired physical function, poor quality of life and reduced survival. As for malnutrition screening, several papers have already reported in patients from Far East Asian countries that the CONUT score is associated with survival in LC patients submitted to surgical treatment [18,19,20,21,22,23,24], while so far much less attention has been paid to LC patients in the advanced stages of disease [16,24,25].

Facing this background, we aimed to assess the risk of malnutrition in patients diagnosed with advanced-stage LC who received different types of medical therapies. The primary objective was to evaluate the effectiveness of the CONUT score as a nutritional screening tool, with the aim of determining its predictive value for short-term radiological response to treatment in the short period and survival outcomes after a significant period of follow up. Moreover, we intended to explore the use of the CONUT score in identifying patients at higher risk of malnutrition.

## 2. Materials and Methods

### 2.1. Study Design and Study Population

This is a single centre, retrospective, observational cohort study conducted in the Clinic of Respiratory Diseases of Monaldi Hospital in Naples. We consecutively enrolled patients with LC admitted to our hospital between January 2020 and September 2024. No patient was involved in the development, design and conducting of this study and the data interpretation.

### 2.2. Inclusion and Exclusion Criteria

The inclusion and exclusion criteria are described in Figure 1.

### 2.3. Study Protocol

The study group consisted of 109 patients. All patients gave their written consent to participate in the research. We obtained all patients’ data from their medical records, including a history and physical examination, contrast-enhanced total body CT, complete blood count and the histological diagnosis. Blood samples were routinely collected at the baseline (at the time of diagnosis, before starting treatment) and after four cycles of treatment (chemotherapy, immunotherapy, chemoimmunotherapy and oral therapy with TKI). Standard biochemical and hematological analyses were performed in the hospital laboratory according to the patient’s normal care path. The CONUT score was calculated based on formulas from existing studies [24] (Table 1). We also performed a cut off value of 2 after a comprehensive review of the literature [22,23,26]. In addition to the CONUT score, we also evaluate other nutritional assessment tools, such as GNRI and PNI.

### 2.4. Histological Data

Data regarding the histopathologic analysis and TNM stage was collected from the patients’ files with metastases diagnosed by computer tomography imaging. The patients were subdivided into two groups: locally advanced (stage III) and advanced (stage IV) disease. Validated questionnaires were used: international physical activity questionnaire (IPAQ), the Mediterranean diet score (MedDietScore) and Montreal Cognitive Assessment (MOCA) [27].

### 2.5. Treatment Response Evaluation

We also evaluated the association between CONUT score and the radiologic response according to Response Evaluation Criteria in Solid Tumours (RECIST criteria) [28], which categorize responses into *Progression Disease* (PD), *Partial Response* (PR) and *Stable Disease* (SD) performed after four cycles of therapy. Moreover, the weight was measured using the same balance and height was measured using a portable stadiometer.

### 2.6. Follow-Up Period

The participants were followed up for a maximum time of five years or until death occurred. The median follow-up period was 30 months. The patients were followed up every month by hospital visits and examination of the patients. The study was conducted in accordance with the Declaration of Helsinki, and the protocol was approved by the Local Department Ethics Committee. Informed consent was obtained from all subjects involved in the study.

### 2.7. Data Collection, Assessment and Follow up

We collected data on demographics, underlying disorders, comorbidity, anthropometry, and laboratory tests at hospital admission and at restaging. Body Mass Index (BMI) was calculated from the time of diagnosis of LC heights and weights of the patients. The 8th edition AJCC (American Joint Committee on Cancer) and the Union for International Cancer Control (UICC) TNM stage classification were used to assess patients’ demographic and clinicopathologic parameters [29]. RECIST guidelines were used to determine the degree of histological response [28]. Follow up on all selected patients was conducted throughout the study period and was performed according to the NCCN guidelines: (1) 3-monthly for the first 1–2 years after surgery; (2) 6-monthly at 3–5 years after surgery; (3) 5-yearly until death. Patients underwent follow ups with regular physical examinations, blood tests and CT total body scan. The follow-up period for the patients was 4 years, starting from the first admission to the Clinic of Respiratory Diseases at Monaldi Hospital. Clinicopathological characteristics and prognosis survey such as Overall Survival (OS) and Progression Free Survival (PFS) were retrospectively analyzed. Observations for each patient were terminated on confirmation of the patient’s death or the end of the observation period. PFS was defined as the interval from treatment initiation until disease progression or death. OS was defined as the interval from treatment initiation until the date of death or the date of last follow-up for patients who had not died before the censor date.

### 2.8. Nutrition-Related Tools

The CONUT score is calculated as follows: CONUT score = serum albumin score + TC score + TLC score. The serum albumin score was: ≥3.5 g/dL = 0; 3.0–3.49 g/dL = 2; 2.50–2.99 g/dL = 4; and <2.50 g/dL = 6. The TLC score was originated as follows: ≥1600/µL = 0; 1200–1599/µL = 1; 2, 800–1199/µL; and 3, <800/µL. The TC score was classified as follows: ≥180 mg/dL = 0; 140–179 mg/dL = 1; 100–139 mg/dL = 2, and <100 mg/dL = 3. Patients were categorized as having no (CONUT = 0–1), mild (CONUT = 2–4), moderate (CONUT = 5–8) or high (CONUT = 9–12) malnutrition risk according to the original study. For this analysis, patients with a CONUT score < 2 were included into the low CONUT group and those with a score ≥ 2 into the high CONUT group (any malnutrition risk). This cut off value was selected based on commonly used thresholds from previous studies.

We also examined other nutritional tools including Geriatric Nutritional Risk Index (GNRI) and Prognostic Nutritional Index (PNI). The GNRI was calculated as follows: GNRI= (14.89 × serum albumin (g/dL)) + (41.7 × actual weight (kg)/ideal weight (kg)). Ideal weight was calculated using the Lorentzian formula. Patients were considered to have no (GnRI ≥ 98), mild (92 ≤ GNRI < 98), moderate (82 ≤ GnRI < 92) or high (GNRI < 82) malnutrition risk according to the original study [30]. The PNI was calculated as follows: PNI= (10 × serum albumin (g/dL)) + (0.005 × TLC (/µL)). Patients were considered to have no (PNI ≥ 50), mild (45 ≤ PNI < 50), moderate (40 ≤ PNI < 45) or high (PNI < 40) malnutrition risk according to a previous study [31].

### 2.9. Statistical Analysis

A preliminary descriptive analysis was performed: summarized data were expressed as medians or numbers with percentages. The patients were divided into two groups based on their nutritional value of CONUT. These two groups were then compared across various factors, including gender, age, ECOG-PS, Histopathology diagnosis, expression of Programmed death ligand 1 (PD-L1), type of treatment, presence of brain metastasis at baseline and nutritional tools. Kaplan–Meier analyses were used to assess PFS and OS and the log-rank test was used to compare the survival distribution between curves. Prognostic factors affecting survival were determined. Hazard ratios (HRs), which represent relative risks, were calculated through Cox regression analysis, along with their corresponding 95% confidence intervals (CIs). Univariate analysis assessed the impact of prognostic factors on OS and PFS. A *p*-value < 0.05 was considered statistically significant. Statistical analysis was performed using Jamovi software (version 2.3) from the Jamovi Project (2023).

## 3. Results

In this study, 109 patients (mean age 66.2 ± 9.6 years, 64.2% men) with NSCLC were enrolled; their main characteristics are presented in Table 2. Eighty patients (73.4%) had a smoking history and 19 of them were currently smokers. More than 70% of the study group had an ECOG-PS between 0 and 1. Seventy-five patients had a histopathologic diagnosis of adenocarcinoma and thirty-four of squamous cell carcinoma, being in stage III (locally advanced, 13.2%) or stage IV (advanced, 86.2%) of disease. Sixty-four patients (58.7%) were classified as having low (0–49%) and 45 patients (41.3%) as having high PD-L1 expression (≥50%). Immunotherapy was administrated as first line treatment to 68 patients: 43 (39.5%) received immunotherapy alone, according to the high expression of PD-L1, while 25 (22.9%) were treated with a combination of chemo and immunotherapy. According to biomolecular profiling 11 patients revealed a positivity for the Epidermal Growth Factor Receptor (EGFR) and Anaplastic lymphoma kinase (ALK) mutations and therefore started oral treatment with TKIs. Thirty patients received only chemotherapy. None of the patients underwent surgical treatment either prior or after enrolment.

CONUT score, GNRI and PNI were used to assess malnutrition risk. CONUT score varied from 0 to 5 with a median value of 2. The high CONUT group (score ≥ 2, i.e., any malnutrition risk) included 32 patients (29.4%), while the low CONUT (0 or 1) group included 77 patients (70.6%). As shown in Table 3, the mean values of both PNI and GNRI were significantly higher in the low CONUT group compared to the high CONUT group. However, there were no significant differences between the two groups in terms of the type of treatment received. In addition, patients in the high CONUT group were more likely to present with brain metastases.

### 3.1. Nutritional Risk and Radiologic Response to Therapy

A major objective of the study was to evaluate the association between CONUT score and the radiologic response to therapy (RECIST criteria [28]). It was found that the LC patients with a low CONUT score had a significantly better response, with higher PR and SD, whereas those with a CONUT score ≥ 2 had a worse response in terms of PD (Table 3).

The association between CONUT score and the radiologic response is shown in Figure 2. Analysis showed that LC patients with progression of disease comparing to those without progression of disease exhibited greater value of CONUT score median of 1.75 (IQR: 1.29–1.75) vs. median of 1.24 (IQR: 1.13–1.24) and lower values of GNRI [101.70 ± 4.50 vs. 103.31 ± 6.50 (*p* = 0.052)] and PNI [51.14 ± 4.27 vs. 52.99 ± 10.12 (*p* < 0.001)].

Concerning the determinants of the radiologic response to therapy, the CONUT score emerged as a significant predictor of a greater risk of disease progression in both crude and adjusted models with an HR around 1.7 (*p* < 0.001) (Table 4). On the contrary, GNRI and PNI did not show significant associations.

### 3.2. Survival Analysis Based on CONUT Score

After a 4 yr follow-up period, 34 LC patients had passed away, 43 were still alive continuing their treatment and 32 had completed treatment and were undergoing clinical and radiological follow up every three months. At the time of analysis, the median follow-up period was 30 months for the entire cohort and 14 months for patients who had passed away. Nine patients in the low CONUT group died from LC-related causes while five died from other diseases. In contrast, in the high CONUT group, fifteen patients died from LC-related causes and five died due to other diseases.

OS and PFS were both greater in patients with CONUT score < 2 than in those with CONUT ≥ 2. Accordingly, the Kaplan–Meier analysis (Figure 3) revealed a significantly worse prognosis (OS and PFS) in the high compared to the low CONUT group. The OS at 36 months was 48.4% and 75.8% in the high and low CONUT group, respectively. A survival analysis was also performed for PNI and GNRI. At 36 months, OS was 63.4% in the high PNI group and 33.2% in the low PNI group (*p* = 0.025, Figure 3), while it was 72.3% and 40.5% in the high and low GNRI groups, respectively (*p* = 0.022), (Figure 3).

Finally, a Cox regression analysis for OS and PFS was performed including several possible predictors (Table 5a,b). In univariable analysis for OS, ECOG-PS 0-1 (HR = 5.53, 95% CI 1.62 to 18.84) or 2 (HR = 4.41, 95% CI 1.20 to 16.20), brain metastases (HR = 3.61, 95% CI 1.76 to 7.42), CONUT score (HR = 1.47, 95% CI 1.16 to 1.86) and radiologic response to treatment in terms of PR (HR = 0.02, 95% CI 0.01 to 0.08) and SD (HR = 0.08, 95% CI 0.04 to 0.19) were significantly associated with OS, while age, stage of disease, GNRI and PNI score were not. Multivariable Cox regression analysis indicated that the CONUT score remained a significant predictor of OS (HR= 1.47 (CI 1.16 to 1.80), *p* = 0.004) along with other prognostic factors such as ECOG PS, BMI and the radiologic response to therapy. Similarly in multivariable analysis, ECOG PS 0-1, CONUT score and radiologic response to treatment emerged as significant predictors of PFS (Table 5b).

## 4. Discussion

There is an increasing interest in the role of malnutrition and inflammation as predictors of clinical outcomes in cancer patients [32,33], with several studies showing significant relationships and less favourable prognosis [34,35]. The present paper shows that malnutrition risk, assessed using the CONUT score, is associated with the effectiveness of treatment (i.e., radiological response according to the RECIST guidelines), and with OS and PFS as well, in NSCLC patients living in a European country.

The CONUT score is calculated based on serum albumin, lymphocyte count and total serum cholesterol level [11,14,24]. Low albumin levels were related to poor outcomes in patients with malignant diseases [36], reflecting both malnutrition and inflammation; it is worth noting that pro-inflammatory cytokines like interleukin 6 (IL-6) and tumour necrosis factor alpha (TNF-α) decrease albumin synthesis [37,38]. In addition, serum cholesterol values are, to some extent, associated with tumour progression; for instance, serum TC and HDL-C were found to be directly related to OS in cancer patients [38,39]. Finally, lymphocytes are crucial for cancer immune surveillance and the destruction of cancer cells through immune mechanisms [40]; low lymphocyte count might suggest a pre-existing state of immunosuppression, which is expected to be linked to a poorer LC prognosis [40]. However, Neutrophil-to-Lymphocyte Ratio (NLR) has emerged as a significant prognostic biomarker in various cancers, including LC, showing consistent correlation with both OS and PFS [41], likely due to its reflection of systemic inflammatory burden and tumour-promoting immune responses [42,43]. Emerging evidence suggests that its prognostic performance may be enhanced when considered in combination with Absolute Lymphocyte Count (ALC) [44].

Concerning the present paper, it should be noted that research has already highlighted the prognostic role of the CONUT score before surgery in LC patients (almost always NSCLC) from Far East Asian Countries [27,28,29]; little information is available on short-term postoperative pulmonary complications [45], while there is more consistent evidence on the association of malnutrition risk with poor survival [21,46]. In addition, little attention has so far been paid to the advanced stages of disease, as well as to patients submitted to chemotherapy, radiotherapy, etc. [16,26,31].

The present study included 109 patients with NSCLC (living in Southern Italy) in locally advanced and advanced stage of disease and submitted to different types of medical therapy [47]. The CONUT score was used in order to identify patients with any malnutrition risk (score ≥ 2 according to what was proposed in the original article [48]), who were 29.4% of the total sample, a figure that is consistent with several previous studies [24]. Indeed, in other studies, that proportion was greater [26], possibly because of the various cut-off values used in the literature (for instance, ≥1 or ≥3 [23,24,31].

As for clinical outcomes, the CONUT score was recently reported to be a predictor of therapeutic effects in a small group of NSCLC patients undergoing treatment with pembrolizumab [23]. The present study extended those findings by including LC patients (advanced stages) who received different types of therapy: thirty patients received chemotherapy, forty-three immunotherapy, twenty-five a combination of chemo-immunotherapy, while eleven were administrated TKIs. The responses were categorized as PD, PR, or SD. After four cycles of therapy, LC patients with a CONUT score < 2 showed significantly better responses, with higher rates of PR and SD, compared to those with a higher CONUT score. Apparently, there was no difference depending upon the type of therapy. In contrast, patients with a CONUT score ≥ 2 exhibited less favourable radiologic outcomes, characterized by a higher incidence of PD. It is worth noting that this finding was also confirmed not only by univariable but also by multivariable Cox regression analysis. Of note, no significant associations were observed for two other screening tools, i.e., GNRI and PNI, raising the hypothesis that CONUT score is more effective in defining the patient’s prognosis.

While several papers have already reported that the CONUT score is associated with OS or PFS in LC patients submitted to surgical treatment [18,19,20,21,22], much less attention has so far been paid to the advanced stages of disease and only in LC patients from Far East Asian countries [16,23,24,25,49]. In this specific case, the CONUT score was found to be a prognostic marker in III-IV NSCLC patients receiving first-line chemotherapy [24] and in those undergoing treatment with pembrolizumab [23]. In patients not submitted to surgery, OS significantly decreased when a higher malnutrition risk was found (using CONUT score or PNI) in both NSCLC and SCLC [16].

Differently from the previous ones, the present study was carried out in a European country, involving LC patients who received immunotherapy, chemotherapy or TKIs. According to our data, the one-year survival rate was significantly higher in the low-CONUT group compared to the high-CONUT group and the difference between the two groups was also confirmed by K-M curve over a longer follow-up period. Furthermore, Cox regression analysis indicated that the CONUT score (≥2) was as a significant determinant of OS and PFS in the patients studied even after considering other possible predictors, thus confirming what has been shown for LC patients submitted to surgical treatment (less advanced stages of disease) [18,21,22,45,46,50,51]. Of note, a more effective prognostic role of CONUT score was suggested by the fact that no significant association with OS or PFS was observed for two other screening tools, i.e., GNRI and PNI. However, Takahashi et al. demonstrated that both GNRI and PNI were significant factors influencing OS and PFS in LC patients in all age groups [20]. Shoji et al. identified that the GNRI was a significant predictor of OS in patients with lung cancer over 75 years old [52]. Furthermore, Soomin An et al. conducted a comparison between GNRI and PNI as nutritional indicators, revealing that GNRI is either superior to or at least on par with PNI as a predictor of OS and PFS in patients with a diagnosis of NSCLC with stage I-III, regardless of age [19].

### Strengths and Limitations of the Present Study

This study presents several strengths. Notably, our study revealed that patients with lower scores in the GNRI and the PNI, as well as higher scores on the CONUT scale, were associated with unfavourable outcomes, including reduced OS and PFS. Therefore, our findings suggest that nutritional status, as indicated by these metrics, plays a significant role in the prognosis for patients, highlighting the importance of monitoring and optimizing nutritional health to potentially improve patient outcomes. Furthermore, this research was carried out by well-trained personnel according to well-defined inclusion criteria and standardized clinical procedures, using a diagnostic threshold derived from the original paper. Nevertheless, some limitations need to be noted. First, it is a single-centre study with a reasonable but not large sample size, and therefore the results cannot be generalized to other groups of patients and should be carefully interpreted while subgroup analysis was not taken into consideration. this study was limited to NSCLC patients. We do not have any data regarding Small Cell Lung Cancer (SCLC). There may be other potential factors influencing the outcomes; actually, due to a lack of information, our study did not account for confounding variables such as socio-economic status and family background, which could influence the results. Therefore, further studies are needed to evaluate the relevance and applicability of the CONUT score in SCLC populations. Further multicentric, prospective studies with larger and more diverse populations are warranted to validate our findings and to explore the clinical relevance of nutritional scores, such as CONUT, GNRI, and PNI, in both NSCLC and SCLC cohorts.

## 5. Conclusions

The CONUT score is a screening tool for malnutrition risk, based on simple hematobiochemical parameters, that is straightforward to use in clinical settings. It has proven to be valuable for predicting survival outcomes in patients with lung cancer. Nutritional interventions based on CONUT score assessment may also be necessary to improve patient outcomes. We recommend that patients calculate their CONUT score regularly before undergoing therapy and during the follow up.

Differently from previous studies, the present paper included LC patients from a European country, adding new information concerning patients with advanced LC submitted to different clinical treatments (chemotherapy, chemotherapy plus immunotherapy or only immunotherapy). indicating an association of malnutrition risk, measured by CONUT score, with the effectiveness of treatment (i.e., radiological response according to the RECIST guidelines) and with OS and PFS as well.

## Figures and Tables

**Figure 1 nutrients-17-03416-f001:**
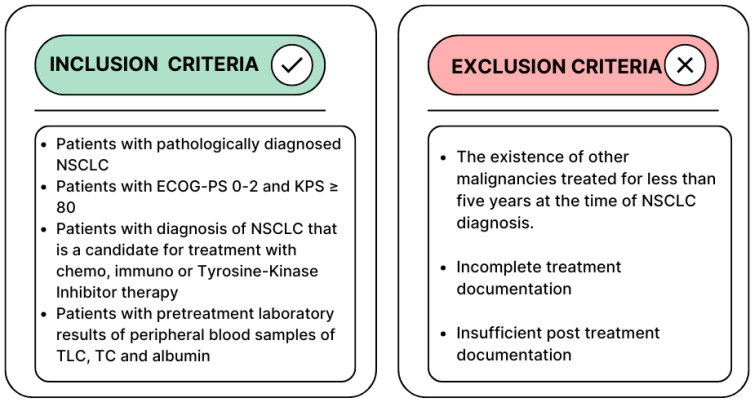
The inclusion and exclusion criteria of the study.

**Figure 2 nutrients-17-03416-f002:**
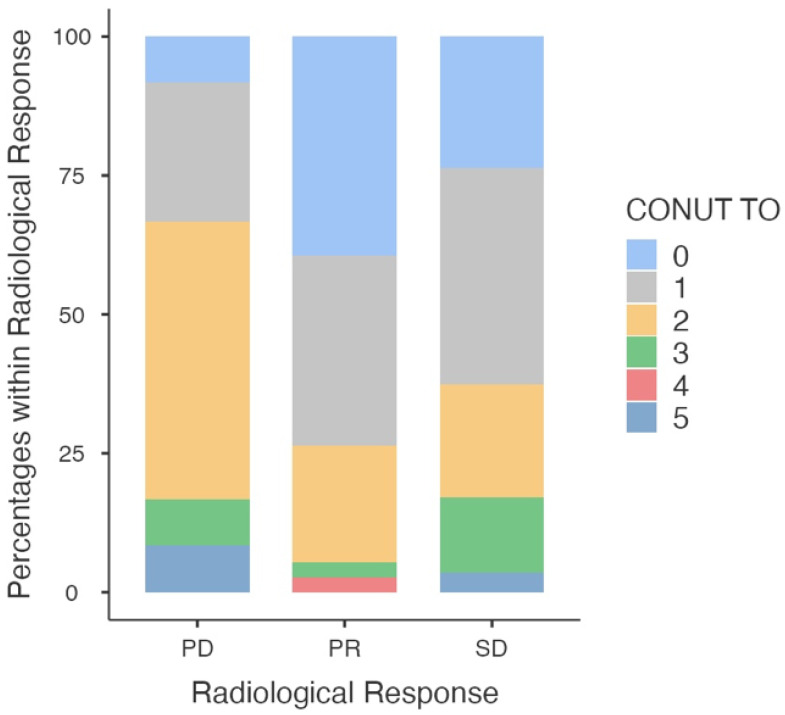
The association between CONUT score and the radiologic response. The x-axis represents the radiologic response categories: PD: progression of disease, PR: partial response; SD: stable disease while the y-axis shows the radiologic response as a percentage.

**Figure 3 nutrients-17-03416-f003:**
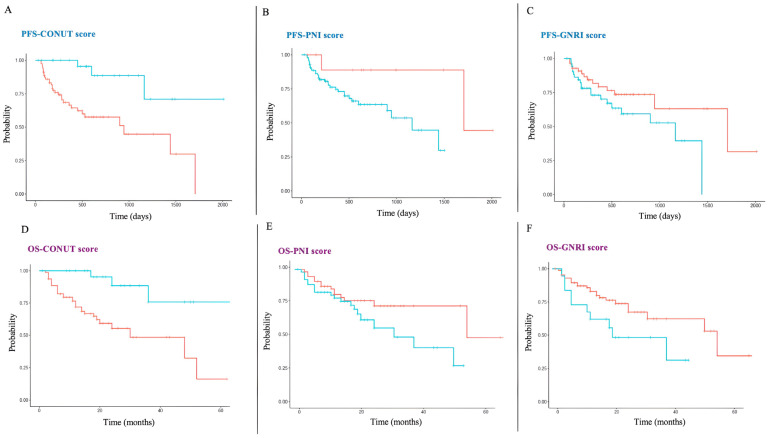
Relationship of PFS and OS with nutritional risk in patients with NSCLC (Kaplan–Meier analysis). (**A**) PFS according to the CONUT score (low and high group). (**B**) PFS according to PNI. (**C**) PFS according to GNRI. (**D**) OS for the CONUT (low and high group) of all patients with NSCLC. (**E**) Kaplan–Meier analysis of OS for the PNI of all patients with NSCLC. (**F**) Kaplan–Meier analysis of OS for the GNRI of all patients with NSCLC. *Red lines*: patients with high malnutrition risk. *Blue lines*: patients with no/low nutritional risk.

**Table 1 nutrients-17-03416-t001:** Scoring system for the controlling nutritional status (CONUT). CONUT score = serum albumin score + total lymphocyte score + total cholesterol score.

Parameters	Normal	Light	Moderate	Severe
**Serum Albumin (g/dL)**	≥3.50	3.00–3.49	2–50–2.99	<2.50
** *Score* **	0	2	4	6
**Total lymphocyte count**	≥1600	1200–1599	800–1199	<800
** *Score* **	0	1	2	3
**Total cholesterol (mg/dL)**	>180	140–180	100–139	<100
** *Score* **	0	1	2	3
**CONUT score (total)**	0–1	2–4	5–8	9–12
** *Assessment* **	Normal	Light	Moderate	Severe

**Table 2 nutrients-17-03416-t002:** Baseline characteristics of the study population.

Variable	Number (%)
Total patients	109
Males	70(64.2%)
Age ± SD, years (median age)	66.2 ± 9.6 years (68)
≥60 years	85 (78%)
Smoking history, *n* (%) Never smoker Ex smoker Current smoker	29 (26.6%) 61 (56%) 19 (17.4%)
ECOG-PS 0 1 2	38 (34.9%) 44 (40.4%) 27 (24.7%)
Histopathology NSCLC, *n* (%) Adenocarcinoma Squamous	75 (68.8%) 34 (31.2%)
Stage Diseases, *n* (%) III IV	15 (13.2%) 94 (86.2%)
Expression of PD-L1 <1% 1–49% ≥50%	14 (12.8%) 50 (45.9%) 45 (41.3%)
Type of treatment, *n* (%) Chemotherapy Immunotherapy Chemo-immunotherapy TKI	30 (27.5%) 43 (39.5%) 25 (22.9%) 11 (10.1%)
Survival status, *n* (%) Exitus Alive	34 (31.2%) 75 (68.8%)

**Table 3 nutrients-17-03416-t003:** The relationship between CONUT score and the clinicopathological characteristics of patients.

	Patients with CONUT Score < 2	Patients with CONUT Score ≥ 2	*p*-Value
Gender (male)	46 (65.7%)	24 (34.3%)	0.775
Age	65.7 (IQR: 23–79)	67.4 (IQR: 49–84)	
ECOG-PS T0			0.109
0	28 (40.6%)	10 (25%)	
1	28 (40.6%)	16 (40%)	
2	13 (18.8%)	14 (35%)	
Histology NSCLC			0.360
squamous	18 (26.1%)	12 (30%)	
adenocarcinoma	51 (73.9%)	28 (67.5%)	
Stage			0.934
III	16 (23.2%)	9 (22.5%)	
IV	53 (76.8%)	31 (77.5%)	
PD-L1 expression			0.436
negative	11 (15.9%)	3 (7.5%)	
weak	30 (43.5%)	20 (50%)	
strong	28 (40.6%)	17 (42.5%)	
Therapy			0.756
single agent immuno	25 (36.2%)	18 (45%)	
chemotherapy	19 (24.6%)	11 (27.5%)	
chemo-immuno	17 (24.6%)	8 (20%)	
TKIs	8 (11.6%)	3 (7.5%)	
Brain Metastasis (YES)	8 (11.6%)	11 (27.5%)	0.035
GNRI	103 ± 7.5	101 ± 5.9	<0.001
PNI	54.0 ± 7.9	50.6 ± 11.9	<0.001
PFS (months)	18.3 ± 14.2	12.1 ± 10.5	<0.05
OS (months)	20.6 ± 13.9	15.2 ± 10.7	0.011
Radiological response after 4 cycles of therapy PR SD PD	28 (25.7%) 37 (34.0%) 6 (5.5%)	10 (9.2%) 22 (20.2%) 6 (5.5%)	<0.001 <0.001 <0.001

IQR: interquartile range; TKI: tyrosine kinase inhibitors; GNRI: Geriatric Nutritional Risk Index; PNI: Prognostic Nutritional Index; PFS: Progression Free Survival; OS: Overall Survival; PR: partial response; SD: stable disease; PD: Progression Disease.

**Table 4 nutrients-17-03416-t004:** Association of malnutrition risk (assessed by CONUT score, GNRI and PNI) with the radiologic response to therapy.

	Crude Model		Adjusted Model *	
	HR (95% CI)	*p*-Value	HR (95% CI)	*p*-Value
CONUT	1.71 (1.36–2.15)	<0.001	1.72 (1.36–2.19)	<0.001
GNRI	1.00 (0.97–1.02)	0.709	1.00 (0.98–1.02)	0.856
PNI	0.97 (0.97–1.02)	0.644	1.00 (0.98–1.03)	0.694

* Adjusted for age > 60 years, ECOG-PS, stage of disease. Not significant for sex, BMI, histology and type of therapy.

**Table 5 nutrients-17-03416-t005:** (**a**) Univariate and multivariate analysis of Overall Survival (OS) in patients with lung cancer. (**b**) Univariable and multivariable analysis of Progression Free Survival (PFS) in patients with lung cancer.

(**a**)				
	**Univariable**		**Multivariable**	
**VARIABLES**	**HR (95% CI)**	** *p* ** **-Value**	**HR (95% CI)**	** *p* ** **-Value**
** *AGE > 60 YO* **	1.03 (0.42–2.51)	0.952	0.70 (0.25–1.95)	0.495
** *ECOG-PS* ** **0–1** **2**	5.53 (1.62–18.84) 6.83 (1.93–24.17)	0.006 0.003	4.41 (1.20–16.20) 4.93 (1.20–20.17)	0.025 0.027
** *STAGE OF DISEASE* ** **III** **IV**	0.91 (0.17–4.66) 1.92 (0.42–8.68)	0.909 0.399	1.03 (0.11–9.91) 0.48 (0.09–2.62)	0.977 0.396
** *BMI* **	0.96 (0.89–1.04)	0.349	0.85 (0.72–1.01)	0.060
** *CONUT* **	1.47 (1.16–1.86)	0.001	1.57 (1.20–1.97)	0.004
** *GNRI* **	0.98 (0.95–1.02)	0.371	1.04 (0.96–1.12)	0.329
** *PNI* **	0.95 (0.89–1.01)	0.106	1.01 (0.97–1.06)	0.628
** *RESPONSE AFTER 4 CYCLES* **				
**PR**	0.02 (0.01–0.08)	<0.001	0.02 (0.01–0.10)	<0.001
**SD**	0.08 (0.04–0.19)	<0.001	0.07 (0.02–0.22)	<0.001
**PD**	0.25 (0.08–0.33)	0.310		
** *BRAIN METASTATIS* **	3.61 (1.76–7.42)	<0.001	1.29 (0.50–3.28)	0.599
(**b**)				
	**Univariable**		**Multivariable**	
**VARIABLES**	**HR (95% CI)**	** *p* ** **-value**	**HR (95% CI)**	** *p* ** **-value**
** *AGE > 60 YO* **	0.96 (0.39–2.36)	0.928	1.12 (0.36–3.50)	0.841
** *ECOG-PS* ** **0–1** **2**	5.55 (1.63–18.97) 6.12 (1.73–21.58)	0.004 0.005	4.38 (1.22–15.77) 3.18 (0.78–13.00)	0.044 0.107
** *STAGE OF DISEASE* ** **III** **IV**	0.95 (0.17–5.18) 2.05 (0.45–9.29)	0.949 0.352	0.89 (0.10–8.29) 0.32 (0.06–1.81)	0.921 0.197
** *BMI* **	0.96 (0.86–1.04)	0.337	0.81 (0.67–0.98)	0.030
** *CONUT SCORE* **	1.45 (1.15–1.83)	<0.001	1.47 (1.16–1.80)	0.004
** *GNRI* **	0.98 (0.95–1.02)	0.358	1.00 (0.95–1.05)	0.920
** *PNI* **	0.95 (0.89–1.01)	0.110	0.98 (0.90–1.06)	0.533
** *RESPONSE AFTER 4 CYCLES* **				
**PR**	0.03 (0.01–0.11)	<0.001	0.04 (0.01–0.22)	<0.001
**SD**	0.13 (0.06–0.28)	<0.001	0.12 (0.04–0.37)	<0.001
**PD**	0.25 (0.08–0.33)	0.310		
** *BRAIN METASTATIS* **	3.74 (1.82–7.69)	<0.001	1.50 (0.56–4.02)	0.420

PD: progression of disease; PR: partial response; SD: stable disease.

## Data Availability

Data are contained within the article.

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
