# Peer review of "Controlling Nutritional Status (CONUT) Score as a Predictor of Prognosis in Non-Small Cell Lung Cancer"

_nutrients, 2025, doi:10.3390/nu17213416_

Round 1

Reviewer 1 Report

Comments and Suggestions for Authors

In the manuscript nutrients-3944529, the authors aimed to reveal the impact of nutritional status on the prognosis of patients with lung cancer. The manuscript is based on 58 refs, most of them recently published. Numerous abbreviations are included in the manuscript; the authors are invited to put a table with all of them and their significance at the end of the manuscript. 

The authors are also invited to diminish the similarities from 29% to under 20%.

Introduction

Please show the Hypotheses for the present study, based on literature data

Materials and Methods

The first subsection (lines 97-137) is an enormous epic paragraph with many aspects, consequently presented. The authors are invited to show these data more concisely. They are encouraged to separate the described data into more subsections, such as Study design, Inclusion/exclusion criteria, Study protocol, Histological data, Response evaluation, Follow-up, and Calculated indexes, for better understanding.

Where is Table 1, mentioned in line 119?

The paragraph from lines 170-178 is subject to Discussion.

 The inclusion/exclusion criteria should be included in a table.

The statistical analysis software (type and version) is missing.

Results

Please revise the phrase from lines 246 - 249, for better understanding.

Please revise this phrase: "There may be other potential Moreover, in our study we did not estimate confounding [...] (line 427).

In each Table footer and figure, please note the abbreviations used for better understanding. 

Please present data from tables and Figures better in the manuscript text. 

Discussions

Please make a different subsection entitled Strengths and Limitations of the present study and complete it with more relevant statements.

Please show if the study hypotheses were verified.

Comments on the Quality of English Language

Moderate revision. 

Author Response

Please find attached our answers.

Reviewer 2 Report

Comments and Suggestions for Authors

Please address the following comments and suggestions

Comments

Line 24. Relationships treatment response and survival. The sentence needs clarity

relationship between treatment response and survival

Line 99. Data recorded timeline Jan 2012-Sept 2023 contradicts as mentioned in line 150 that patients were followed up for 4 years. Please check

Line 68, 88. Unnecessary citations. Please revise all references and remove irrelevant references

Line 118. Please add reference to validate CONUT cut off value of 2

Line 479, 481. Reference 11 and 12 both are same. Please check

p-values inconsistency such as p < 0.05, p=0.033. Please check through out of manuscript

Plagiarism is high especially in methodology section. Please reduce the plagiarism.

Suggestions

The title of manuscript can be improved to make it more meaningful

Such as “Prognostic Impact; Controlling Nutritional Status (CONUT) Score”

Author Response

Please find attached our answers.

Reviewer 3 Report

Comments and Suggestions for Authors

Dear authors,
I appreciate your excellent research. I only have a few minor comments that could help improve the manuscript:
1.    On line 298, a closing parenthesis is missing; please correct it.
2.    Why was the Neutrophil-to-Lymphocyte Ratio not calculated? This marker has been shown to be a significant prognostic factor in various cancer types, both for Overall Survival and Disease-Free Survival, and is also related to inflammation and tumor aggressiveness. Several studies suggest that combining with Absolute Lymphocyte Count can improve prognostic accuracy. It is recommended that this indicator be incorporated into the discussion and provided for further consideration.
3.    It is recommended that you review the text, as there is additional white space on several lines.
4.    In Figure 2, it would be beneficial to increase the font size, as the graph legends are currently challenging to read.
5.    Linea 490: The quotation marks must be removed from the author's name.

Author Response

Please find attached our answers.

Round 2

Reviewer 2 Report

Comments and Suggestions for Authors

Thank you for addressing the comments and suggestions and incorporating the revisions into the manuscript